# An Optimum Tea Fermentation Detection Model Based on Deep Convolutional Neural Networks

**Gibson Kimutai [1,\*], Alexander Ngenzi [1], Rutabayiro Ngoga Said [1], Ambrose Kiprop [2,3] and Anna Förster [4]**

[1]   African Center of Excellence in Internet of Things (ACEIoT), College of Science and Technology, University of Rwanda, P.O. Box, 3900 Kigali, Rwanda; yngenzi37@gmail.com (A.N.); said.rutabayiro.ngoga@gmail.com (R.N.S.)

[2]   Department of Chemistry and Biochemistry, Moi University, P.O. Box, 3900-30100 Eldoret, Kenya; profakiprop@gmail.com

[3]   African Center of Excellence in Phytochemicals, Textile and Renewable Energy (ACE II-PTRE), P.O. Box, 3900-30100 Eldoret, Kenya

[4]   Sustainable Communication Networks, University of Bremen, 8359 Bremen, Germany; anna.foerster@comnets.uni-bremen.de

\*   Correspondence: kimutaigibs@gmail.com

**Abstract:** Tea is one of the most popular beverages in the world, and its processing involves a number of steps which includes fermentation. Tea fermentation is the most important step in determining the quality of tea. Currently, optimum fermentation of tea is detected by tasters using any of the following methods: monitoring change in color of tea as fermentation progresses and tasting and smelling the tea as fermentation progresses. These manual methods are not accurate. Consequently, they lead to a compromise in the quality of tea. This study proposes a deep learning model dubbed TeaNet based on Convolution Neural Networks (CNN). The input data to TeaNet are images from the tea Fermentation and Labelme datasets. We compared the performance of TeaNet with other standard machine learning techniques: Random Forest (RF), K-Nearest Neighbor (KNN), Decision Tree (DT), Support Vector Machine (SVM), Linear Discriminant Analysis (LDA), and Naive Bayes (NB). TeaNet was more superior in the classification tasks compared to the other machine learning techniques. However, we will confirm the stability of TeaNet in the classification tasks in our future studies when we deploy it in a tea factory in Kenya. The research also released a tea fermentation dataset that is available for use by the community.

**Keywords:** machine learning; deep learning; image processing; classification; tea; fermentation

## 1. Introduction

Tea is one of the most popular and lowest cost beverages in the world [1]. Currently, more than 3 billion cups of tea are consumed every day worldwide. This popularity is attributed to its health benefits, which include prevention of breast cancer [2], skin cancer [3], colon cancer [4], neurodegenerative complication [5], prostate cancer [6], and many others. Tea is also attributed to the prevention of diabetes and boosting metabolism [7]. Depending on the manufacturing technique, it may be described as green, black, oolong, white, yellow, and compressed tea [8]. Black tea accounts for approximately 70% of tea produced worldwide. The top four tea-producing countries are China, Sri Lanka, Kenya, and India (Table 1).

**Table 1.** Top tea-producing countries globally.

| Rank | Country | Percentage |
|:---:|:---:|:---:|
| 1 | China | 20.6% |
| 2 | Sri Lanka | 19.3% |
| 3 | Kenya | 18.2% |
| 4 | India | 7.5% |

Kenya is the largest producer of black tea in the world [7] due to its low altitude, rich loamy soil conditions, ample rainfall, and a unique climate [9]. In Kenya, tea is produced by small- and large-scale farmers. Small-scale farmers are more than 562,000 and account for about 62% of the total tea produced in Kenya [10]. The rest are produced by large-scale tea plantations that operate 39 factories. Smallholder farmers are managed by the government through the Kenya tea development agency (KTDA) board [11]. The board manages 66 tea factories across the country where smallholder tea is processed [1]. Tea is regarded as a significant contributor to the country's economy as it is the leading exchange earner and contributes to more than 4% of the gross domestic product (GDP). The sector is also a source of livelihoods to more than 10% of the country's estimated population of 40 million people [12,13]. Despite the importance of tea to the country, the sector is facing a myriad of challenges which include high production cost, mismanagement, bad agricultural practices, climate change, market competition from other countries, low prices, and lack of automation, among others [13].

There are 5 steps in the production of black tea (Figure 1). The process starts with the plucking of green tea, where two leaves and a bud is the standard. The next step is withering, where tea leaves are spread on a withering bed for them to lose moisture.

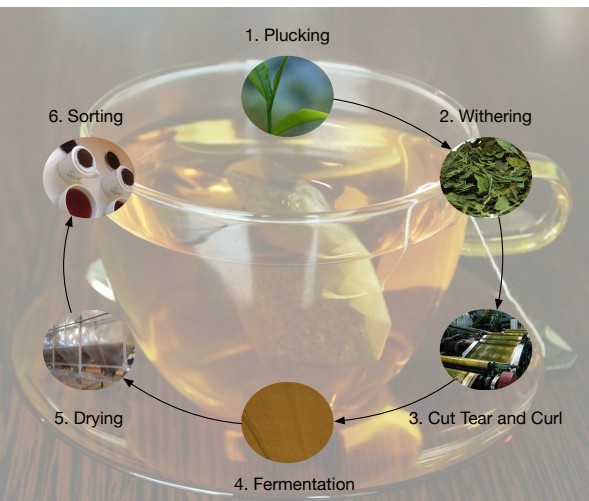

**Figure 1.** Processing steps of black tea.

There is then the cut, tear, and curl step, where tea leaves are cut and torn to open them up for oxidation. The fermentation stage is where tea reacts with oxygen to produce compounds that are responsible for the quality of tea. Heat is passed through tea in the drying stage to remove moisture. The last step is sorting where tea is put into various categories based on their quality. Out of these steps, fermentation is the most important in determining the quality of tea produced [14].

The fermentation process begins when cells of ruptured tea leaves react with oxygen to produce two compounds: Theaflavins (TF) and Thearubins (TR) [15,16]. Theaflavins are responsible for the brightness and briskness of the tea liquor while TR is responsible for the color, taste, and body of tea [16]. During fermentation, the following parameters must be maintained: temperature, relative humidity, and time [15]. The optimum temperature under which fermentation should take place should be approximately 25 °C. The ideal humidity should be approximately 42% [17]. Fermentation

is a time-bound process (Figure 2); at the beginning, the liquor is raw and with a green infusion. The formation of TF and TR increases with time until optimum fermentation is achieved. At the optimum fermentation time, the liquor is mellow and with a bright infusion. This is the desired point in fermentation. After optimum fermentation time, the formation of TR reduces and degradation of TF begins. This stage is over-fermentation, where the liquor is soft and with a dark infusion.

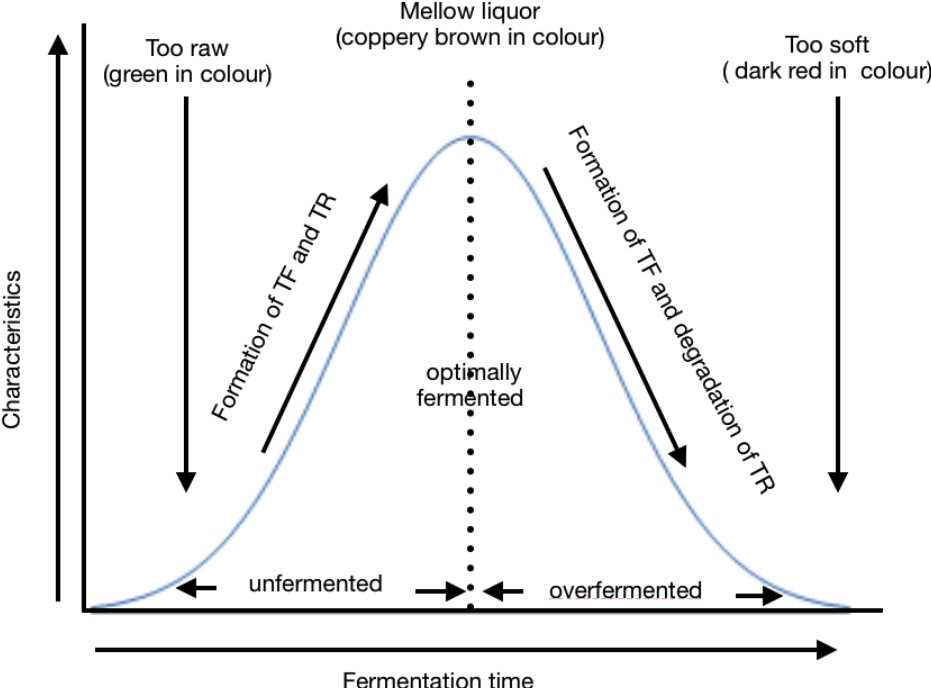

**Figure 2.** Tea fermentation process.

Currently, tea tasters determine optimum fermentation manually by either of the following methods: smell peaks, color change, infusion, or tasting of tea. The constant intervention of humans in a fermentation room disturbs the environment created for fermentation and is also unhygienic. Moreover, humans are subjective and prone to error [7]. These manual methods lead to a compromise in the quality of produced tea and translate to low prices of tea. Therefore, there is a need for alternative means of monitoring the process of fermentation which is the focus of this research.

Currently, machine learning has been applied to many different fields: engineering, science, education, medicine, business, accounting, finance, marketing, economics, stock market, and law, among others [18–22]. Machine Learning (ML) is a branch of artificial intelligence (AI) that enables a system to learn from concepts and knowledge [23]. Deep learning is a collection of machine learning algorithms which models high-level abstractions in data with nonlinear transformations [24]. Deep learning works with the principle of the Artificial Neural Networks (ANN) system, and its fundamental computation unit is a neuron [19,24,25]. In ML, feature extraction and classification are in different steps, while in deep learning, they are in a single step and are done concurrently.

The contribution of this paper is twofold: First, this research proposes a deep learning model based on CNN for monitoring black tea during fermentation. Secondly, this research releases a tea fermentation dataset [26]. The rest of the paper is arranged as follows: presentation of some of the studies aimed at digitizing fermentation is done in Section 2 and a discussion of materials and methods used in this research is presented in Section 3. We provide implementation of the models and the evaluation metrics in Section 4, while Section 5 provides the results and their discussions. We conclude the study in Section 6.

## 2. Related Work

With advancements in computing, digitization across many fields is being witnessed [27]. In agriculture, tea processing has been receiving attention from researchers. Proposals have been made on improving the detection of optimum fermentation using the following techniques: electronic nose, electronic tongue, and machine vision. An electronic nose is a smart instrument designed to detect and discriminate odors using sensors [28]. The basic elements of an electronic nose are an odor sensor and an actuator. Proposals to use the electronic nose in detecting optimum tea fermentation have been proposed in References [14,29–32]. In Reference [31], a handheld electronic nose is proposed, while in Reference [32], ultra-low power techniques have been incorporated into an electronic nose. From the literature, it is evident that the electronic nose has made technological breakthroughs. However, they have not been implemented in many tea factories due to its high price and since they are power-hungry. Going into the future, adoption of the electronic nose will depend on innovative ways of using low-cost sensors in their design. It will also depend on the ability to apply ultra-low-power power design techniques to minimize power consumption.

Some studies exist on the application of an electronic tongue to monitor tea fermentation. They include References [33–36]. In Reference [33], an optimal fermentation model with the use of electronic nose and machine learning techniques is proposed. In Reference [34], the authors applied CNN in the development of an electronic tongue. In Reference [37], an electronic tongue to monitor biochemical changes during tea fermentation is proposed. The authors in Reference [35] designed an electronic tongue for testing the quality of fruits. Research in Reference [36] proposes an electronic tongue with the use of a KNN algorithm and adaptive boosting for development. A fusion of the electronic nose and electronic tongue technologies has been proposed in Reference [38]. It is evident from the literature that there have been proposals to use the electronic tongue in detecting optimal fermentation. However, they have not been implemented in tea factories because these technologies are power-hungry and expensive.

The rapid development of computer vision technology in recent years has led to an increased usage of computational image processing and recognition methods. Proposals to apply image processing in the fermentation of tea are reported in References [39–44]. Research in Reference [39] proposes a quality indexing model for black tea during fermentation using image processing techniques. Another remarkable research is in Reference [40], which detects changes in color during fermentation. In Reference [41], artificial neural networks (ANN) and image processing techniques are applied to detect color changes of tea during fermentation. Research in Reference [42] applied SVM with image processing to detect optimum tea fermentation. In Reference [43], the authors used image processing to detect the color change of tea during fermentation. The authors in Reference [44] implemented an electronic tongue with machine vision to predict the optimum fermentation of black tea. From the literature, tea fermentation is an active research area with authors suggesting different approaches. However, the tea fermentation dataset has not to be used. The use of image processing is the most viable approach due to the low cost of imaging devices. Additionally, a color change is easy to detect compared to taste and odor.

## 3. Materials and Methods

After acquiring data, the next phase was data preprocessing where activities discussed in Section 3.2 were done. The cleaned data was fed to the ML classifiers for training (Figure 3). The training involved hyperparameter tuning until the models were fully trained. Some of the hyperparameters are the learning rate, number of the epoch, regularization coefficient, and batch size. Currently, the available optimization strategies include grid search, random search, hill-climbing, and bayesian optimization, among others [45]. In this study, we adopted the grid search and random search methods. The models were then validated and evaluated using the data discussed in Section 3.1.

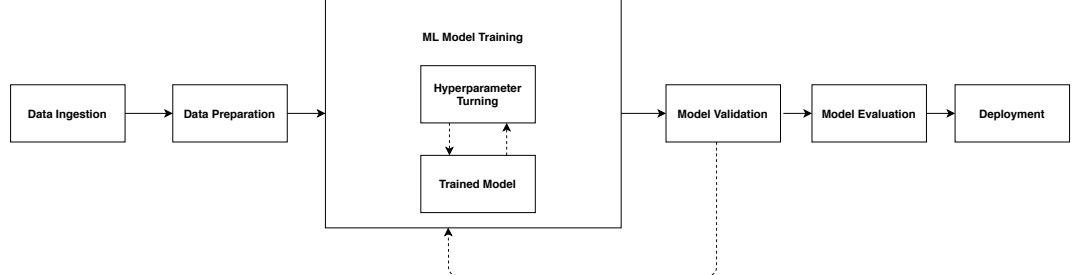

**Figure 3.** Implementation of machine learning techniques.

In model validation, models which did not pass the validation tests were taken back to the training phase. The evaluation results are presented in Section 5. The models can be deployed to a tea fermentation environment after the aforementioned steps.

This section discusses the datasets used, data preprocessing steps, feature extraction methods, machine learning classification models, and the proposed deep learning model.

### 3.1. Datasets

In this paper, two datasets were used: tea fermentation and LabelMe datasets. Since there was no existing standard dataset on tea fermentation images existing in the community, we used the LabelMe dataset to validate our results for it is widely used by researchers in image classification to report their results, the dataset is available at no cost, and there was no available dataset on images of tea fermentation images. We discuss each of the datasets in the following paragraphs.

### 3.1.1. Tea Fermentation Dataset

The images in the tea fermentation dataset [26] were taken in a black tea fermentation environment in a tea factory in Kenya. We used a 5-megapixel camera connected to a Raspberry Pi 3 model B+ to capture the images. Fermentation dataset contains 6000 images that were captured during the fermentation of black tea. Figure 4 shows an image of each of the classes of the tea fermentation dataset. The classes of the images in this dataset are: underfermented, fermented, and overfermented.

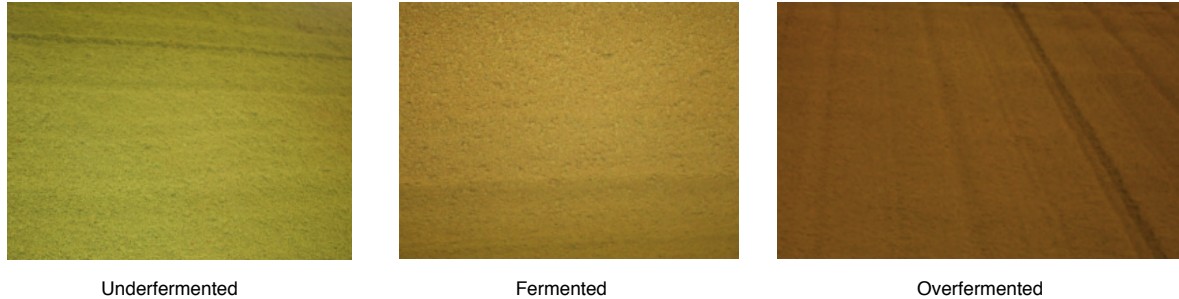

|　　　Underfermented　　　|　　　Fermented　　　|　　　Overfermented　　　|

**Figure 4.** Examples of classes of the tea fermentation dataset.

Table 2 shows the number of images for every class that was used as training, validation, and testing datasets for the classification algorithms. The 80/20 ratio of training/test data is the most commonly used ratio in neural network applications and was adopted in this research. Besides, 10% subset of the test dataset was used to validate the results. A total of 4800 images distributed equally to the 3 classes of images were used for training of the models. To perform validation, 40 images were used in each of the classes while 360 images were used to test the model in each of the 3 classes.

**Table 2.** The image dataset comprising of three classes of images.

| Class | Images Used for Training | Images Used for Validation | Images Used for Testing |
|---|---|---|---|
| Underfermented | 1600 | 40 | 360 |
| Fermented | 1600 | 40 | 360 |
| Overfermented | 1600 | 40 | 360 |
| Total | 4800 | 120 | 1080 |

### 3.1.2. LabelMe Dataset

As explained in Section 3.1, the other dataset that we adopted in this study is the LabelMe dataset [46]. The dataset is one of the standard datasets which researches in the field of image classification use to report their results. The dataset contains 2688 images from 3 classes of outdoor scenes. The classes are forest, coast, and highway. Examples of images from each of the classes are shown in Figure 5.

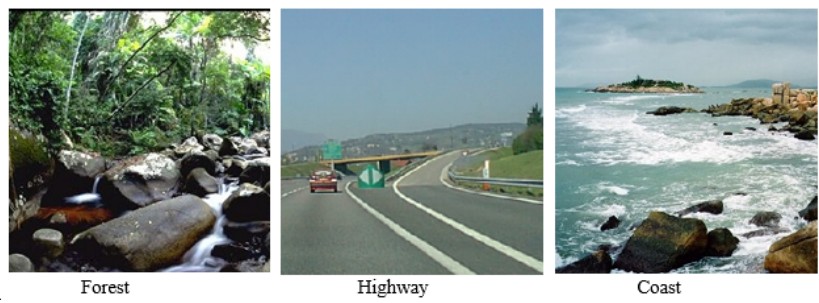

Forest      Highway      Coast

**Figure 5.** Examples of categories of LabelMe dataset.

Table 3 shows the number of images used for training, validation, and testing in each of the categories. As with the case in Section 3.1.1, we adopted the 80/20 ratio for training and testing and 10% for validation.

**Table 3.** Number of images used for training, validation, and testing in the LabelMe dataset.

| Class | Images Used for Training | Images Used for Validation | Images Used for Testing |
|---|---|---|---|
| Coast | 717 | 18 | 161 |
| Forest | 717 | 18 | 161 |
| Highway | 717 | 18 | 161 |
| Total | 2151 | 54 | 483 |

### 3.2. Data Preprocessing and Augmentation

After collecting the images as discussed in Section 3.1.1, the images were resized to $150 \times 150$. Resizing images to $150 \times 150$ before inputting them into different networks was done to adapt different pretraining CNN structures. We adopted the semantic segmentation annotation method discussed in Reference [47] to annotate the images. There are numerous types of noise in images but the most common are photon noise, readout noise, and dark noise [48,49]. To perform denoising, we adopted the linear filtering method.

### 3.3. Feature Extraction

Feature extraction in image processing is the process of extracting image features. It is the most crucial step in image classification as it directly affects perfomance of the classifiers [50]. There are various techniques of feature extraction, but in this paper, we adopted color histogram for color feature extraction and Local Binary Patterns (LBP) algorithm for texture extraction.

### 3.3.1. Color Feature Extraction

Color is an important feature descriptor of an image. During tea fermentation, the color change is evident as the process continues. Relative color histograms in different color spaces can be used to describe tea fermentation images. There are several color spaces which include Red-green Blue (RGB), Hue Saturation Value (HSV), and Hue Saturation Brightness (HSB), among others [51–54]. RGB color space represents a mixture of red, green, and blue. This is the color space that was used to represent the images in this paper. We used color histogram [55] to extract color features of the images that are then fed to the classifiers for training, evaluation, and testing. To construct a feature vector from the color histogram, we used OpenCV [56]. The input was an image of RGB color space. The RGB color space was converted to HSV and represented by 3 channels (the hue, the saturation, and the value). We used 8 bins to represent the three channels. Finally, the range of the channels was between 0–150 since the images had been resized to 150 by 150 pixels. Figure 6a shows an image of underfermented tea, while Figure 6b shows the corresponding color histogram.

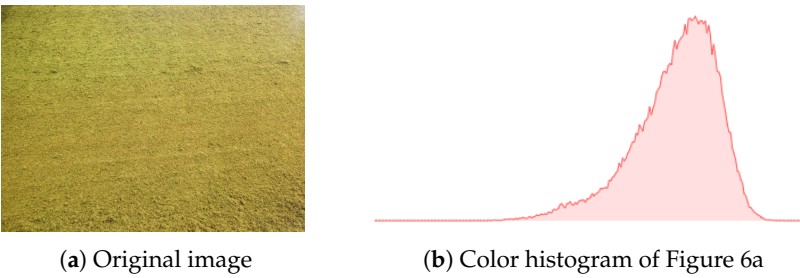

(**a**) Original image          (**b**) Color histogram of Figure 6a

**Figure 6.** Generation of color features of an image using color histogram.

### 3.3.2. Texture Feature Extraction

Textures are characteristic intensity variations that originate from the roughness of an object surface. The texture of an image is classified into first-order, second-order, and higher-order statistics [57]. There are a variety of methods of extracting texture features including Local Binary Patterns (LBP), the Canny edge detection, discrete wavelet transform, and gray level occurrence matrix, among others [58–60]. In this paper, we adopted LBP to extract the texture features of the images. LBP has many advantages which include reduced histograms and consideration of the center pixels point effect [61], among others. The LBP algorithm is represented by Equation (1):

$$LBP_{x_c,y_c} = \sum_{n=0}^{7} 2^n (I_n - I(x_c, y_c)) \tag{1}$$

where $LBP_{x_c,y_c}$ is the value at the center pixel $x_c, y_c$, $I_n$ is the values of neighbor pixel, and $I(x_c, y_c)$ is the intensity at the center pixel.

The steps of the texture feature extraction were as follows:

1. The original image was converted into a grayscale image using the approach discussed in Reference [62]. The color grayscale image generated is shown in Figure 7b.
2. LBP algorithm was then used to calculate each of the pixels in the grayscale image as shown in Figure 7. Both $LBP_{x_c,y_c}$ value and texture image are generated. The generated texture image is shown in Figure 7c.
3. Finally, the texture image obtained was converted into gray- scale histogram as shown in Figure 7d.

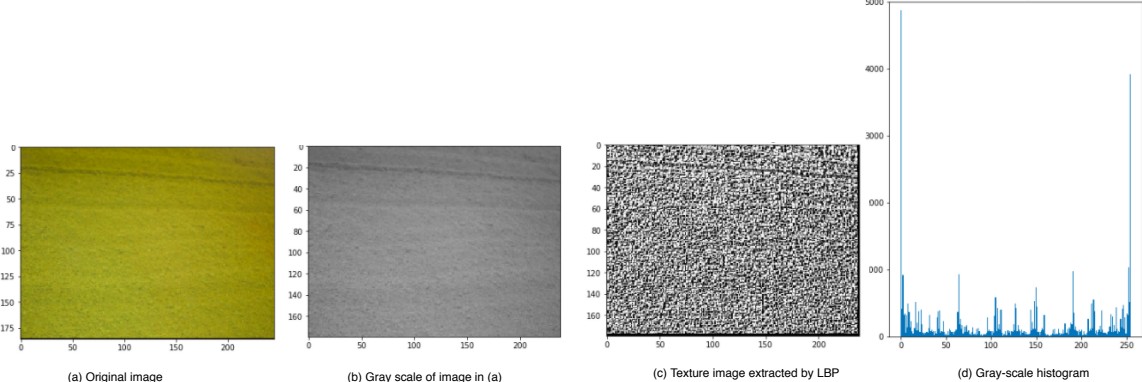

(a) Original image    (b) Gray scale of image in (a)    (c) Texture image extracted by LBP    (d) Gray-scale histogram

**Figure 7.** Conversion of image to grayscale histogram using Local Binary Patterns (LBP).

### 3.4. Classification Models

In this paper, we perform the classification of the images in the datasets discussed in Section 3.1 using the following classifiers: decision tree, random forest, K- nearest neighbor, TeaNet, support vector machine, linear discriminant analysis, and naive Bayes. The next paragraphs discuss each of the classifiers.

### 3.4.1. Decision Tree (DT)

Decision tree is a machine learning technique that employs a tree structure to specify the order of the decisions and the consequences [63]. During training, it generates rules and decision trees. The generated DTs are followed in the classification of the new data [64]. It has the following constituents: root node, internal node, and leaf node (Figure 8). Branches and leaves point to the factors that concern a particular situation [65].

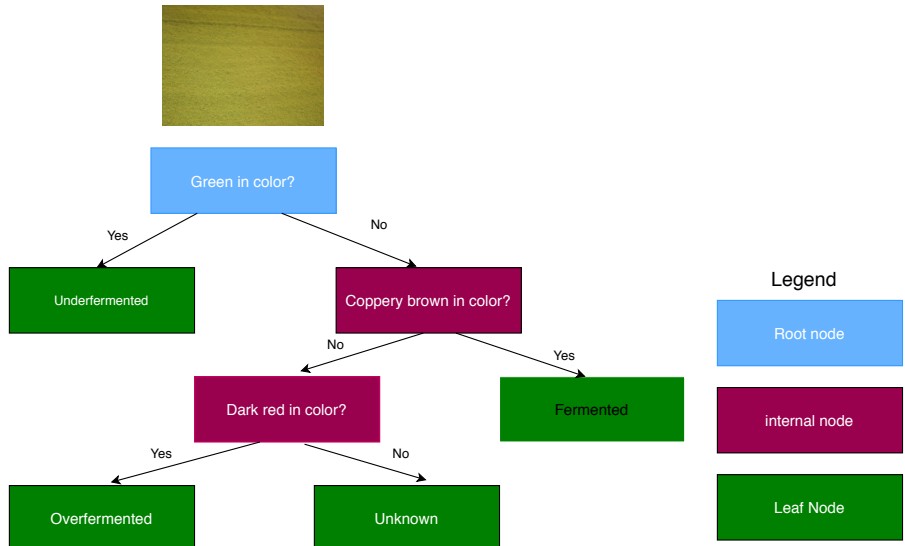

**Figure 8.** Example of classification by a decision tree.

It is one of the most used machine learning algorithms in classification [66,67] because of its advantages which include high tolerance to multicollinearity [68], flexibility, and exclusion of factors which are not important automatically [63,69], among others. However, DT training is relatively expensive as complexity and time taken are more, a small change in data changes the DT structure, and it is inadequate in predicting continuous values, among others [70].

### 3.4.2. Random Forest (RF)

Random forest is a machine learning model that operates by constructing multiple decision trees during training [71,72]. The constructed multiple trees are then used for prediction during classification. Each individual tree in the random forest outputs a class prediction and the class with most votes becomes the model's prediction [73]. Figure 9 shows an example of classifying an image using random forest. The image was classified as belonging to class A since the majority of the trees (2) classified the image as belonging to class A. The classifier can estimate missing data, can balance errors in datasets where classes are imbalanced, and can be used for both classification and regression [74–77]. Additionally, it has better results compared to decision tree algorithm; the random forest has a better classification result [72,78]. However, random forest is not as effective in regression tasks as it is in classification and is a black box model [79–81].

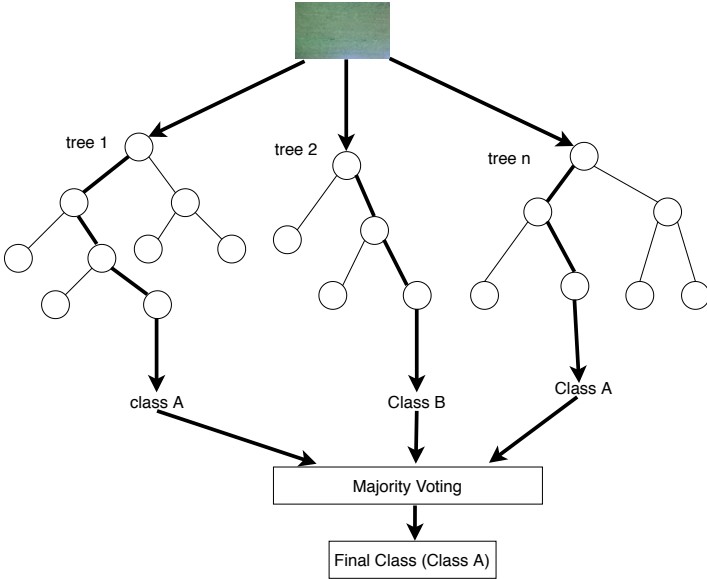

**Figure 9.** Example of a random forest operation.

### 3.4.3. K-Nearest Neigbor (KNN)

K-Nearest Neighbor (KNN) is a nonparametric machine learning model used for classification and regression [82–84]. In classification tasks, KNN determines the class of a new sample based on the class of its nearest neighbors. During decision making in a classification task, it finds k training instances that are closest to the unknown instance. It then picks the most occurring classification for the k instances [85]. It determines a dominant category to the target object in which k is the number of training samples. This algorithm assumes that samples close to each other belong to the same category in classification [84]. Figure 10 illustrates an example of a classification using KNN. The task is to find a class that the triangle belongs to. It can either belong to the blue ball class or the green rectangle class. The k is the algorithm we wish to take a vote from. In this case, let us say k = 4. Hence, we will make a cycle with the triangle as the center just to enclose only three data points on the plane. Clearly, the triangle belongs to the blue ball class since all of its nearest neighbors belong to that class.

The algorithm is simple to implement and has a robust search space [86–88]. The main challenge of the model is the expense incurred in terms of large computations in identifying neighbors in a large amount of data [89,90].

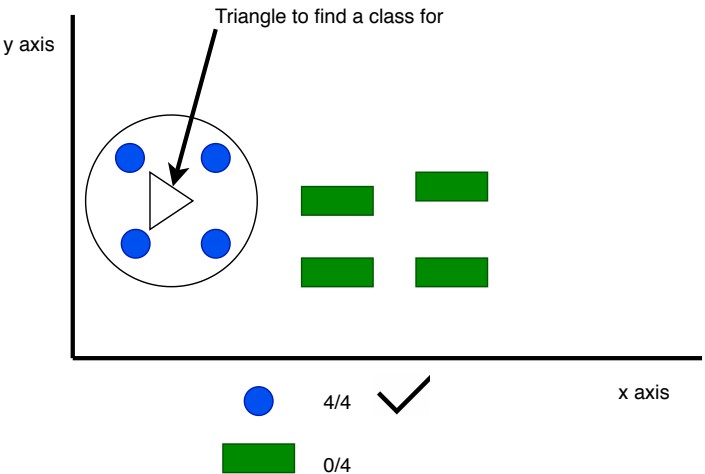

**Figure 10.** K-Nearest Neighbor (KNN) proximity algorithm map.

3.4.4. Convolutional Neural Network (CNN)

Convolutional Neural Network (CNN) is a class of deep learning technique that is currently emerging in solving computer vision challenges, which includes detection of objects [91], segmentation [92], and dimage classification [93], among others. CNN emerged in the mid-2000s due to the development in computing power of hardware of the computer [94]. A CNN is composed of the following layers (Figure 11): an input layer, convolutional layer, pooling layer, dense layer, and output layer. An input layer of a CNN is the layer where the input is passed to the network. In Figure 11, the input layer contains an image which needs to be classified [95]. Convolutional layers are a set of filters needed to learn. The filters are used to calculate output feature maps, with all units in a feature map sharing the same weights [96–98]. A pooling layer will then sum up the activities and selects the maximum values in the neighborhood of each feature map [99]. A dense layer consists of neurons in a neural network which receive inputs from all the neurons in the previous layer [100]. Convolutional has shown high accuracy in image recognition tasks; however, they have high computation tasks [101].

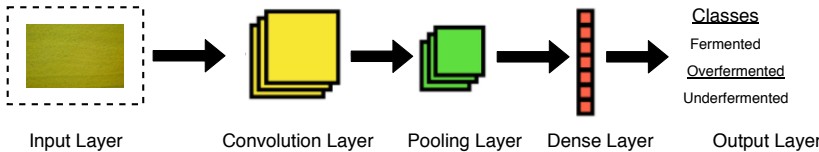

**Figure 11.** A typical Convolution Neural Network (CNN) architecture.

3.4.5. Support Vector Machine (SVM)

Support Vector Machine (SVM) is a non-probabilistic binary classifier that aims at finding a hyperplane with a maximum margin to separate high dimension classes by focusing on the training samples located at the edge of the class distribution [102]. The model is based on statistical learning theory and the structural risk minimization theory [103]. The model chooses extreme vectors which help in creating the hyperplane. These extreme points are referred to as support vectors. The binary classification problem with linear separability (Figure 12) has a goal to find the optimum hyperplane, through maximizing the margin and through minimizing the classification error between each class.

Some of the advantages of SVM is its ability to rely on its own memory efficiency and its ability to work well with classes having distinct margins [104,105]. However, SVM tends to take a large training time for a large dataset and is not effective for overlapping classes [106,107].

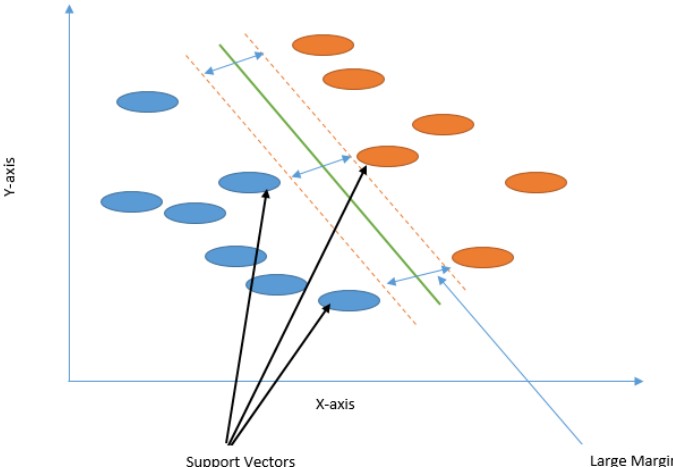

**Figure 12.** An example of a classification task using Support Vector Machine (SVM).

3.4.6. Naive Bayes (NB)

Naive Bayes is a probabilistic model based on Bayes' theorem. Bayes' theorem provides the relationship between the probabilities of two events and their conditional probabilities [108–110]. A Naive Bayes classifier assumes that the presence or absence of a particular feature of a class is unrelated to the presence or absence of other features [111,112]. In classification tasks, NB constructs a probabilistic model of the features and applies the model in prediction of the new instances. Figure 13 shows a sample of balls belonging to two classes: yellow and green. The task is to estimate the class for which the ball with a question mark belongs to. There is a very high probability that the ball belongs to class green since most of the balls belong to that class.

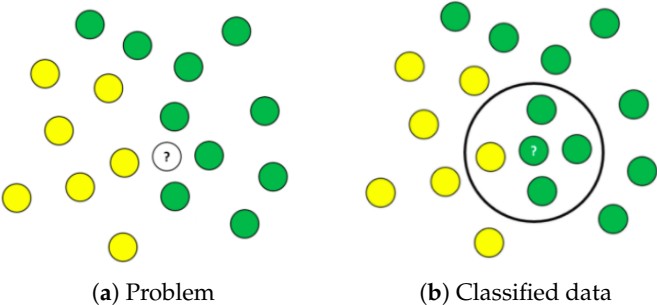

(**a**) Problem       (**b**) Classified data

**Figure 13.** Example of classification using Naive Bayes.

3.4.7. Linear Discriminant Analysis (LDA)

Linear discriminant analysis is an approach developed by the famous statistician R.A. Fisher, who arrived at linear discriminants from a different perspective [113]. He was interested in finding a linear projection for data that maximizes the variance between classes relative to the variance for data from the same class [114]. LDA combines features of a class and builds on separating the classes. It models the differences between classes and builds a vector for differentiating the classes based on the difference in the classes [115,116]. LDA is popular because of its low-cost implementation and its ease of adaptation for discriminating nonlinearly separable classes through the kernel trick method [117], among others. Some of the weaknesses of LDA includes its challenge in handling large datasets, among others [118]. Figure 14a shows a classification problem, while Figure 14b shows the solution to the classification problem using LDA.

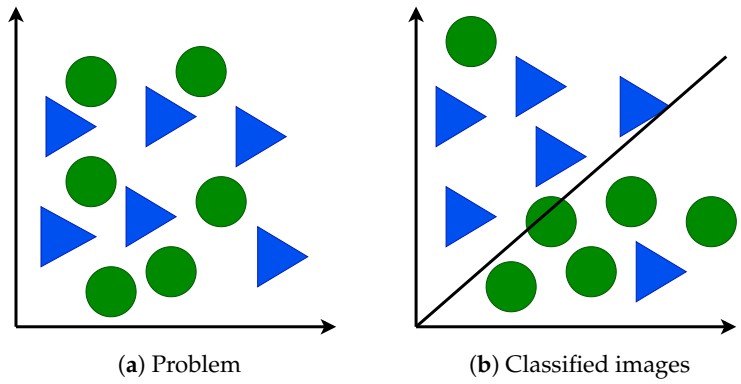

(**a**) Problem   (**b**) Classified images

**Figure 14.** Example of classification using Local Discriminant Analysis (LDA).

### 3.5. TeaNet

TeaNet is a deep learning model based on Convolutional Neural Network (CNN). The network architecture of TeaNet is an improvement upon the standard AlexNet model [119]. We designed an optimum tea fermentation detection model with relatively simple network structure and small computational needs. To construct TeaNet, we reduced the number of convolutional layer filters and the number of nodes in the fully connected layer. This reduces the number of parameters that require training, thus reducing the overfitting problem. The basic architecture of the network is shown in Figure 15.

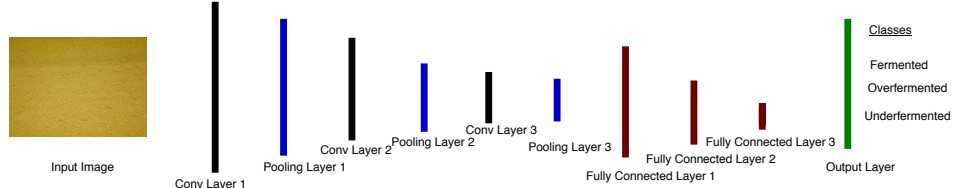

**Figure 15.** The architecture of the TeaNet that we propose for optimum detection of tea fermentation.

The input images were rescaled to 150 × 150 pixels, and the three color channels discussed in Section 3.3.1 were all processed directly by the network. Table 4 shows the layer parameters of TeaNet.

**Table 4.** Layer parameters for TeaNet.

| Layer | Parameter | Activation Function |
|---|---|---|
| input | 150 × 150 × 3 | — |
| Convolution1 (Conv1) | 32 convolution filters (11 × 11), 4 stride | ReLU |
| Pooling1 (Pool1) | Max pooling (3 × 3) 2 stride | — |
| Convolution2 (Conv2) | 64 convolution filters (3 × 3), 1 stride | ReLU |
| Pooling2 (Pool2) | Max pooling (2 × 2) 2 stride | — |
| Convolution3 (Conv3) | 128 convolution filters (3 × 3), 3 stride | ReLU |
| Pooling3 (Pool3) | Max pooling (2 × 2) 2 stride | — |
| Full Connect4 (fc4) | 512 nodes, 1 stride | ReLU |
| Full Connect5 (fc5) | 128 nodes, 1 stride | ReLU |
| Full Connect5 (fc6) | 3 nodes, 1 stride | ReLU |
| Output | 1 node | Softmax |

The layers are defined as follows:

1.  The first convolutional layer comprises of 32 filters and a kernel size of 11 × 11 pixels. This layer is followed by a rectified linear unit (ReLU) operation. ReLU is an activation function that provides a solution to vanishing gradients [96]. Its pooling layer has a kernel size of 3 ×3 pixels, with two strides.
2.  The second convolutional layer comprises of 64 filters and a kernel size of 3 × 3 pixels and is followed by a ReLU operation; its pooling layer has a kernel size of 2 × 2 pixels.
3.  Additionally, the third convolutional layer comprises of 128 filters and a kernel size of 3 × 3 pixels, followed by ReLU with a kernel size of 2 × 2 pixels.
4.  The first full connection layer was made up of 512 neurons, followed by a ReLu and a dropout operation. The dropout operation [120] is proposed to solve overfitting as it trains only a randomly selected nodes. We set the ratio of dropout to 0.5.
5.  The second full convolutional layer had 128 neurons and was followed by a ReLU and dropout operations.
6.  The last fully convolutional layer contains three neurons, which represent 3 classes of images in tea fermentation and LabelMe datasets. The output of this layer is transferred to the output layer to determine the class of the input image. A softmax activation function is then implemented to force the sum of the output values to be equal to 1.0. Softmax also limits the individual output values between 0–1.

At the beginning, the weights of the layers were initialized with random values from a Gaussian distribution. To train the network, a stochastic gradient descent (SGD) technique with a batch size of 16 and a momentum value of 0.9 [121] was adopted. Initially, the learning rate across the network was set to 0.1, and a minumum threshold was set at 0.0001. The number of epochs was set as 50, and the weight decay was set to 0.0005. The accuracy of TeaNet increased with an increase in epoch, and it achieved an accuracy of 1.0 at epoch 10 (Figure 16a). At the beginning of the iteration, the accuracy is low since the weights of the neurons are not fully set. After each iteration, the weights are updated. The validation accuracy shows a steady increase, and the model had an accuracy of 1.0. The loss of TeaNet during training and validation is illustrated in Figure 16b. There is a steady reduction in the loss from the first epoch up to epoch 10, where the loss value is at 0 for both the training and validation sets. From Figure 16, the model has good performance and is not overfitted as it records good results in unseen data.

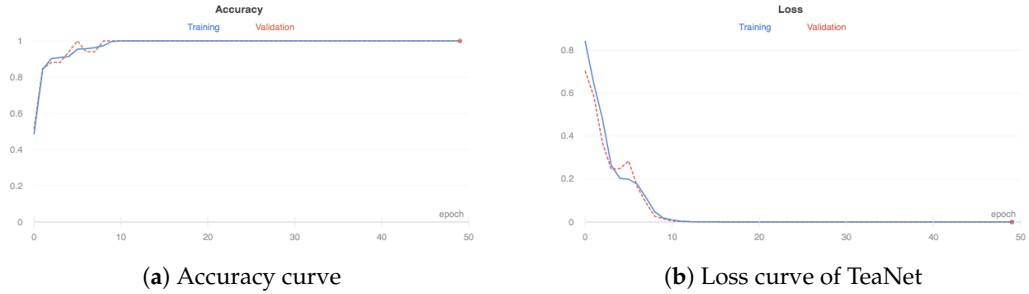

(**a**) Accuracy curve          (**b**) Loss curve of TeaNet

**Figure 16.** Accuracy and loss of TeaNet during training and validation.

## 4. Implementation

To implement the classification models discussed in Section 3.4 and the TeaNet model discussed in Section 3.5, python programming language was adopted. After implementation, it was necessary to evaluate the performance of the various classification models. This section provides the implementation of the models and the metrics that were adopted in evaluating their performances.

*4.1. Implementation of the Classifiers*

As mentioned in Section 4, we adopted python programming language to implement the classification models. Some of the reasons for adopting python were that it has rich libraries [122], that it has moderate learning curve [123], and that it is free and open source [124]. Some of the libraries that we adopted alongside python are Tensorflow [125,126], Keras [127], Seaborn [128], matplotlib [129], sklearn [130,131], OpenCV [56], pandas [132], and numpy [133]. The implemented classification models are available at http://classifier.sisiboportal.com/.

*4.2. Evaluation Metrics*

In Section 4, we mentioned the various evaluation metrics that were adopted in this study to evaluate the classification models. The following paragraphs discuss each of the metric in detail.

4.2.1. Precision

Precision is the ratio of the correct classification to the total number of classifications [134,135]. A low precision indicates a large number of false positives [136]. It can be represented by Equation (2).

$$Precision = \frac{TP}{TP + FP} \tag{2}$$

where *TP* is an outcome where the model correctly classifies a class and *FP* is an outcome where the model incorrectly classifies a class.

4.2.2. Recall

Recall is the ratio of the number of correctly classified images to the total number of images [134,136]. It is the actual positives that are correctly classified to the correct classes. Recall can be represented by Equation (3).

$$Recall = \frac{TP}{TP + FN} \tag{3}$$

where *TP* is an outcome where the model correctly classifies the positive class and *FN* is an outcome where the model incorrectly classifies the negative class.

4.2.3. F1-Score

F1 Score is the harmonic mean between precision and recall. It tells how precise a classifier is in the classification tasks as well as how robust it is [137]. It is represented by Equation (4).

$$F1 - Score = 2 \times \frac{P \times R}{P + R} \tag{4}$$

where *P* is the precision and *R* is the recall.

4.2.4. Accuracy

Accuracy is the fraction of predictions that the model got right. Therefore, it is the sum of correct predictions divided by all the predictions. It can be represented by Equation (5).

$$Accuracy = \frac{TP + TN}{TP + TN + FP + FN} \tag{5}$$

where *TP* is an outcome where a model correctly classifies the positive class, *FP* is an outcome where a model incorrectly classifies the positive class, *TN* is an outcome where the model correctly classifies a negative class, and *FN* is an outcome where a model incorrectly classifies a negative class.

### 4.2.5. Logarithmic Loss

Logarithmic loss or log loss works by penalizing false classifications [134]. In classification tasks, it is the measure of the inaccuracy of classification. An ideal logarithmic loss should be 0. Logarithmic loss can be represented by Equation (6).

$$Loss = -(g(log(p)) + (1 - g)log(1 - p)) \tag{6}$$

where $g$ is the predicted probability and $p$ is the true label.

### 4.2.6. Confusion Matrix

A confusion matrix is used to summarize the classification performance of a classifier with test data. Sensitivity in a confusion matrix measures the proportion of actual positives that are correctly identified and can be represented by Equation (7).

$$Sensitivity = \frac{TP}{TP + FN} \tag{7}$$

where $TP$ is the number of correct classification while $FN$ is an outcome where the model incorrectly classifies the negative class.

## 5. Evaluation Results

In this section, we provide the results of the evaluation of the classification models based on the metrics discussed in Section 4.2.

Results of the precision of the classifiers in the two datasets are shown in Figure 17. All the other classifiers generally categorized the majority of the images correctly. TeaNet classifier clearly performed better than the rest of the classifiers. TeaNet achieved average precisions of 1.0 and 0.96 in the tea fermentation and LabelMe datasets, respectively. Generally, the majority of the classifiers except decision tree produced better precision in the fermentation dataset compared to the LabelMe dataset. This is because there was a distinctive change in color in the 3 categories of fermentation images. The classifiers recorded an average precision of between 0.78–1.00 in the fermentation dataset and between 0.65–0.96 for the LabelMe dataset.

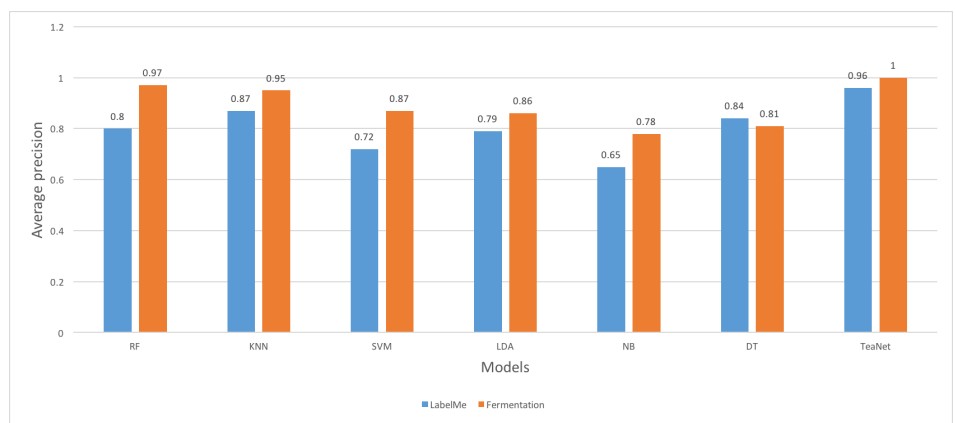

**Figure 17.** Precision of classification for each of the classifiers for the two datasets.

Recall values are illustrated in Figure 18. Once again, TeaNet outperformed the other classifiers by producing the highest average recall values across the datasets. The majority of the classifiers had better performances in the tea fermentation dataset compared to the LabelMe dataset. The classifiers had an average recall of 0.75–1.0 for the tea fermentation dataset and an average of 0.58–0.96 for LabelMe dataset. KNN also had a good performance by recording average recalls of 0.93 and 0.85 for

the tea fermentation and the LabelMe dataset, respectively. Naive Bayes recorded the lowest recall values. From these results, it is evident that TeaNet and KNN produced the best recall values.

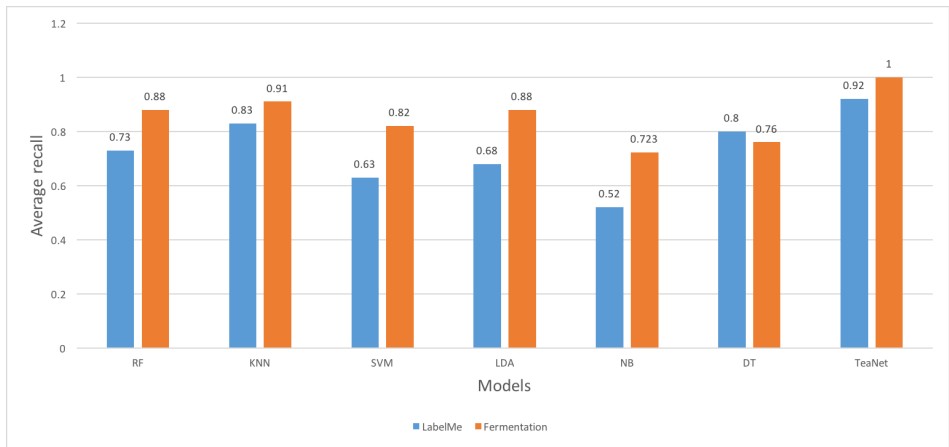

**Figure 18.** Recall of classification for each of the classifiers for the two datasets.

We compared the F1 score of TeaNet with the other classifiers and presented the results in Figure 19. The F1 score values of TeaNet was higher than the other classifiers. The classifiers recorded F1 values of between 0.58–0.9 for the LabelMe dataset and between 0.75–1.00 for the tea fermentation dataset. We can note that TeaNet showed alot of effectiveness as it achieved an F1 of 1.00 in the tea Fermentation dataset and of 0.9 for the LabelMe dataset (Figure 19). KNN also recorded good performances of 0.93 and 0.85 for tea fermentation and LabelMe datasets, respectively.

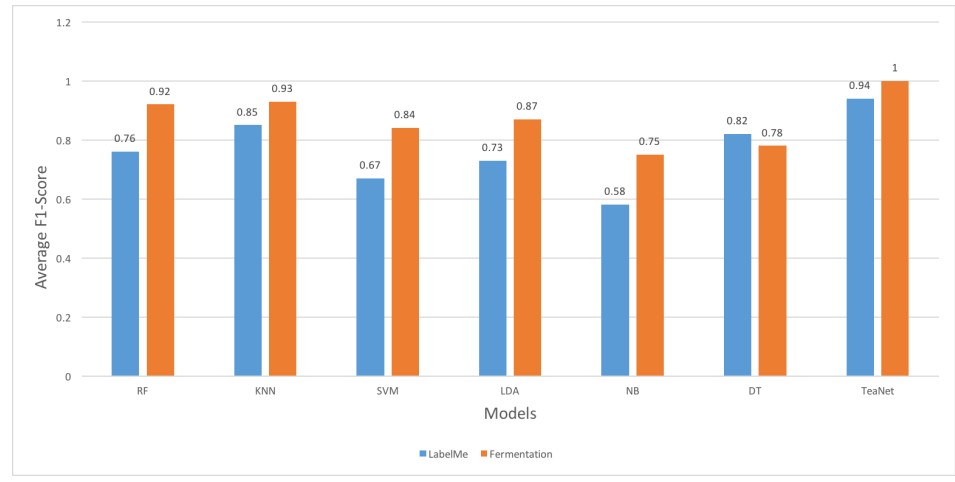

**Figure 19.** F1 scores of classification for each of the classifiers for the two datasets.

The performance of the classifiers in terms of accuracy is presented in Figure 20. The majority of the classifiers had good accuracy results. TeaNet achieved an average accuracy of 1.00 for the tea fermentation dataset and an average accuracy of 0.958 for the LabelMe dataset. This shows that TeaNet once again outperforms the other classifiers. Each of the classifiers produced an accuracy of more than 0.6 across the datasets. It shows that the probability of each of the classifiers in classifying the dataset is more than 60%. Naive Bayes recorded average accuracies of 0.67 and 0.77 for the LabelMe and tea fermentation dataset, respectively. On the other hand, decision tree recorded average accuracies of 0.94 and 0.85 for the tea fermentation and the LabelMe datasets, respectively. These results show that the majority of the classifiers can be applied to detect the tea fermentation images.

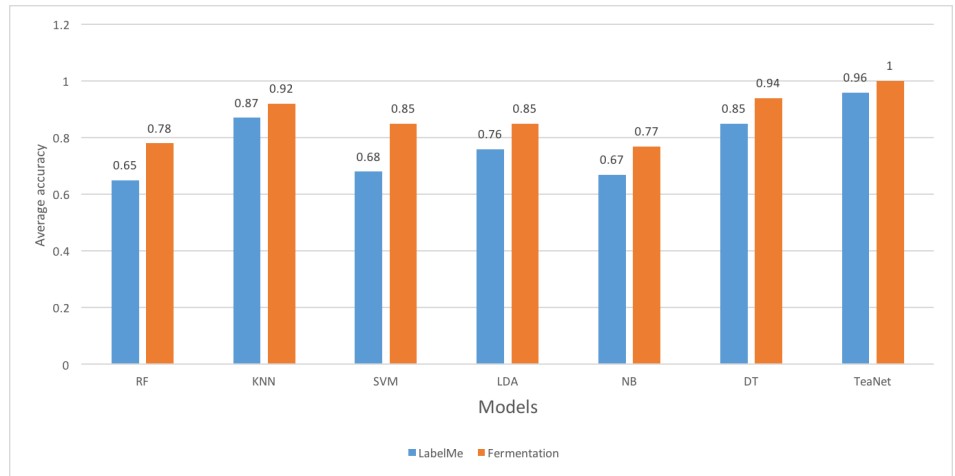

**Figure 20.** Accuracy of classification for each of the classifiers for the two datasets.

Additionally, logarithmic losses of the classifiers were evaluated. The results of the analysis are shown in Figure 21. TeaNet had the least logarithmic loss at 0.136 and 0.09 for LabelMe and Fermentation dataset, respectively (Figure 21). Generally, the Logarithmic losses recorded by the majority of the models was higher than 0.55 for the LabelMe dataset. For tea fermentation, the majority of the models had logarithmic loss of less than 0.50. Evidently, most of the classifiers had a lower logarithmic loss in the fermentation dataset compared to the LabelMe dataset. Random forest recorded the highest logarithmic loss at 0.7 for the LabelMe dataset. On the other hand, Naive Bayes recorded the highest logarithmic loss of 0.64 for the tea fermentation dataset.

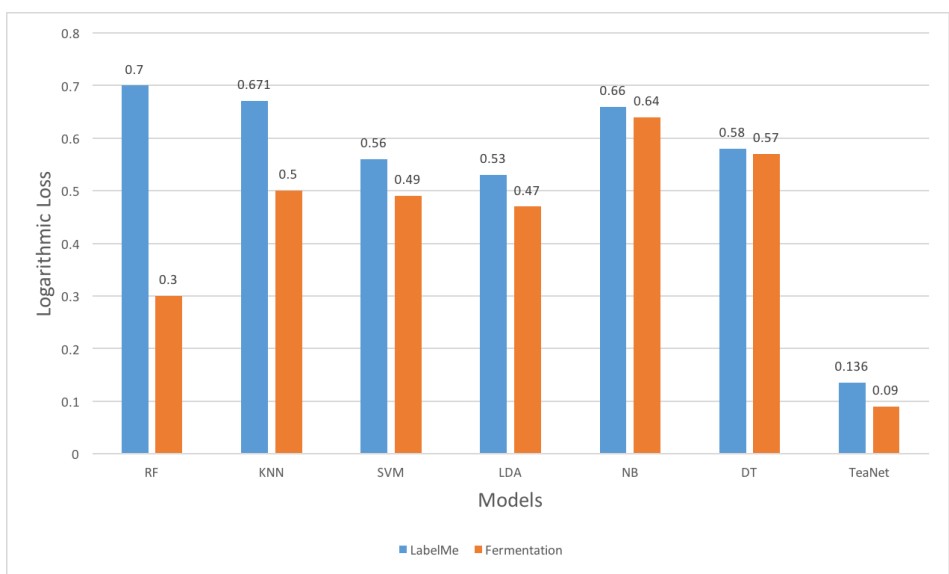

**Figure 21.** Logarithmic Loss of classification for each of the classifiers for the two datasets.

Finally, a confusion matrix was used to further evaluate the classification models and results are shown in Table 5. The least specificity recorded by the classifiers was 73.5%, and the highest was 100.0%. TeaNet recorded an average sensitivity of 100% for fermented, an average of 100% for overfermented, and finally an average of 100% for underfermented. TeaNet outperformed the other classifiers. Consequently, the TeaNet proposed to classify tea images is superior to the other previously described algorithms.

**Table 5.** Confusion matrix of the classifiers.

| Class | Fermented | Overfermented | Underfermented | Sensitivity |
|---|---|---|---|---|
| DT (fermented) | 250 | 32 | 78 | 69.4% |
| DT (overfermented) | 59 | 301 | 0 | 83.6% |
| DT (underfermented) | 271 | 0 | 89 | 75.3% |
| SVM (fermented) | 296 | 22 | 39 | 82.2% |
| SVM (overfermented) | 68 | 291 | 1 | 80.8% |
| SVM (underfermented) | 61 | 0 | 299 | 83.1% |
| KNN (fermented) | 339 | 14 | 7 | 94.2% |
| KNN (overfermented) | 41 | 300 | 19 | 83.3% |
| KNN (underfermented) | 17 | 0 | 343 | 95.3% |
| LDA (fermented) | 331 | 11 | 18 | 92.0 % |
| LDA (overfermented) | 17 | 335 | 8 | 93.3% |
| LDA (underfermented) | 76 | 0 | 284 | 78.9% |
| RF (fermented) | 325 | 14 | 21 | 90.3% |
| RF (overfermented) | 50 | 310 | 0 | 86.1% |
| RF (underfermented) | 45 | 0 | 315 | 87.5% |
| NB (fermented) | 261 | 19 | 80 | 72.5% |
| NB (overfermented) | 89 | 253 | 19 | 70.3% |
| NB (under fermented) | 96 | 0 | 264 | 73.3% |
| **TeaNet (fermented)** | **360** | **0** | **0** | **100.0%** |
| **TeaNet (overfermented)** | **0** | **360** | **0** | **100.0%** |
| **TeaNet (underfermented)** | **0** | **0** | **360** | **100.0%** |

## 6. Conclusions and Future Work

In this paper, we have proposed a deep learning model dubbed TeaNet. We have assessed the capabilities of TeaNet and other standard machine learning classifiers in categorizing images. We used tea fermentation and LabelMe datasets for training and evaluating the classifiers. From the experimental results, TeaNet outperformed the other classifiers in the classification tasks. In general, all the classifiers had good performance across the two datasets. These results show that the majority of the classifiers can be used in real deployments. Importantly, the effectiveness of TeaNet in the tea fermentation dataset is a great achievement. This is a game changer in the application of deep learning in agriculture and most specifically in tea fermentation.

Additionally, the results from this study highlight the feasibility of applying TeaNet in the detection of tea fermentation, which would significantly improve the process. This will, in turn, increase the quality of produced tea and subsequently increase the value of the made tea. This will lead to improved livelihoods of the farmers and to general improvement of the country's GDP. The same technique can be applied to the fermentation of coffee and cocoa. In our future studies, we will implement TeaNet with the Internet of things in real deployment in a tea factory to monitor fermentation of black tea.

**Author Contributions:** Formal analysis, A.F.; methodology, G.K. and A.F.; software, G.K.; supervision, A.N., R.N.S., A.K., and A.F.; validation, G.K.; visualization, G.K.; writing—original draft, G.K.; writing—review and editing, A.N., R.N.S., A.K., and A.F. All authors have read and agreed to the published version of the manuscript.

**Funding:** This research was funded by African Center of Excellence in Internet of Things (ACEIoT) and Kenya Education Network (KENET) through the Computer Science, Information Systems, and Engineering multidisciplinary mini-grant.

**Acknowledgments:** G.K. wishes to thank A.F. for leading this research, for all the support in ensuring this work is a reality, and for the numerous advises and guidance throughout the research process. G.K. also wishes to put on record appreciation to R.N. and A.N. for their inputs in this work. Additionally, G.K. wishes to express appreciation to A.K. for all the mentorship, support, and guidance. Finally, the authors would like to express their appreciation to the anonymous reviewers and editors for their constructive comments.

**Conflicts of Interest:** The authors declare that there is no conflict of interest. The funders had no role in the design of the study; in the collection, analyses, or interpretation of data; in the writing of the manuscript; or in the decision to publish the results

## Abbreviations

ANN       Artificial Neural Network
CNN       Convolutional Neural Network
SVM       Support Vector Machine
DT        Decision Tree
RF        Random Forests
SVM       Support Vector Machine
KNN       K-Nearest Neighbor
OpenCV    Open Computer Vision
IoT       Internet of Things
TF        Theaflavins
TR        Thearubins
EN        electronic Nose
ET        electronic Tongue
LDA       Local Discriminant Analysis
NB        Naive Bayes
KTDA      Kenya Tea Development Agency
GDP       Gross Domestic Product
MSE       Mean Squarred Error
MAE       Mean Absolute Error

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
