# Peer review of "An Optimum Tea Fermentation Detection Model Based on Deep Convolutional Neural Networks"

_data, 2020_

Round 1

Reviewer 1 Report

       This manuscript proposes a CNN model for tea fermentation detection and compares it with six conventional machine learning methods. Experiment results show that CNN achieved better performance than benchmark models. The application of deep learning method in tea production is interesting and practical, which can improve the production efficiency. However, there still exist some questions in the manuscript. Below are my comments. I hope the authors find them useful.

  1. The main goal of the study is tea fermentation detection. Why we use the LabelMe dataset that contains outdoor scenes information in the work?
  2. Figure 4: I am confused about the “Retraining” step in the flowchart.
  3. Figure 17: I think it is better to show the accuracy and loss curves of training and validation sets.
  4. How do you determine the hyperparameters of the machine learning models (e.g., SVM, RF and LDA)? Moreover, Is the TeaNet a successful network applied in other fields? If no, how to determine the CNN network structure? Is there a standard guideline to select the optimal hyperparameters of CNN?
  5. 18-22: The abbreviations of model names is recommended in the figures.
  6. There are a few problems in English language usage.

Reviewer 2 Report

The work is presented in a very readable way. However, I suggest a careful review to:

  • correct a few typos (highlighted in the commented pdf),
  • revise definitions of key metrics for evaluate models (recall-sensitivity) in a multi class classification scenario,
  • a critic discussion over results pretending perfect accuracy and precision while the confusion matrix shows imperfect classification,
  • correct the references to avoid duplicates, complete data and discard too old and irrelevant references.

Please, see the commented PDF.

Reviewer 3 Report

This manuscript proposes a method based on a deep neural network for detecting optimum tea fermentation. The topic is interesting, but the current form of this manuscript is not adequate for the publication. I recommend the following comments to improve the quality and completeness of the manuscript.

(1) In section 3.1.1, the fermentation dataset is composed of 6,000 images, but the total number of images in Table 1 is 6,600. Although this section describes that 10% of the test dataset was used for validation (line 132), the numbers in Table 1 are incorrect. Authors have to describe the information of training, validation and test sets more clearly. There is a similar problem in section 3.1.2.

(2) In section 3.3.1, color features are extracted using a histogram. More details are needed to understand how the authors constructed a feature vector from the color histogram. For example, descriptions about the number of bins and range are required.

(3) In section 3.5, some sentences have to be modified. The original sentences are as follows.
“TeaNet is a deep learning model based on Convolutional Neural Network (CNN). The model 266 is made up of three fully connected layers and a final classification layer. The number of neurons in 267 the first and the second fully connected layers are 512 and 128 neurons respectively. The number of 268 neurons in the third fully connected layer is 3.”
The information on convolutional layers is omitted in this paragraph.

(4) Typos. in section 3.5, Modify “full convolution layer” (line 276-278) to “fully convolutional layer.”

(5) In section 5, the orders of experimental results are needed to be modified. Because the proposed method in this manuscript is TeaNet, placing its result at the end of Table 4 (and at the rightmost place in Figure 18-22) to emphasize the superiority of the proposed method.

Overall, in my opinion, the novelty of this work is rather weak, and a major revision is required for the publication.

Round 2

Reviewer 2 Report

The paper has been improved with to the previous version. I suggest minor changes remarked into the PDF file.

Author Response

Dear Reviewer 2,

Once again, we would like to thank you for your comments which have improved our manuscript further. We have addressed the comments given and marked them as red. Please refer to the following lines on the manuscript: Line: 8, 20, Table 1, Line: 65, 75-76, 117, 120-121, 125, 137, 176, 195-196, 206-208, 215, 309, Figures 17-21 and line 410.

Reviewer 3 Report

The authors made a substantial effort to revise the manuscript according to the reviews’ comments. All of the concerns were resolved, and I believe that this manuscript satisfies the minimum qualifications for the publication.

Author Response

Dear Reviewer 3,

We would like to thank you for reviewing our paper and offering good suggestions which have made our manuscript better. We wish you all the best.